

# Assessing the capability of Fourier transform infrared spectroscopy in tandem with chemometric analysis for predicting poultry meat spoilage

Ubaid ur Rahman[1], Amna Sahar[2], Imran Pasha[1], Sajjad ur Rahman[3] and Anum Ishaq[1]

[1] National Institute of Food Science and Technology, University of Agriculture Faisalabad, Faisalabad, Pakistan
[2] Department of Food Engineering, National Institute of Food Science and Technology, University of Agriculture Faisalabad, Faisalabad, Pakistan
[3] Institute of Microbiology, University of Agriculture Faisalabad, Faisalabad, Punjab, Pakistan

Corresponding author
Amna Sahar, amnasahar@gmail.com

## ABSTRACT

**Background**. Use of traditional methods for determining meat spoilage is quite laborious and time consuming. Therefore, alternative approaches are needed that can predict the spoilage of meat in a rapid, non-invasive and more elaborative way. In this regard, the spectroscopic techniques have shown their potential for predicting the microbial spoilage of meat-based products. Consequently, the present work was aimed to demonstrate the competence of Fourier transform infrared spectroscopy (FTIR) to detect spoilage in chicken fillets stored under aerobic refrigerated conditions.

**Methods**. This study was conducted under controlled randomized design (CRD). Chicken samples were stored for 8 days at $4 + 0.5\,^{\circ}\mathrm{C}$ and FTIR spectra were collected at regular intervals (after every 2 days) directly from the sample surface using attenuated total reflectance during the study period. Additionally, total plate count (TPC), *Entetobacteriaceae* count, pH, CTn (Color transmittance number) color analysis, TVBN (total volatile basic nitrogen) contents, and shear force values were also measured through traditional approaches. FTIR spectral data were interpreted through principal component analysis (PCA) and partial least square (PLS) regression and compared with results of traditional methods for precise estimation of spoilage.

**Results**. Results of TPC ($3.04$–$8.20$ CFU/cm$^2$), *Entetobacteriaceae* counts ($2.39$–$6.33$ CFU/cm$^2$), pH ($4.65$–$7.05$), color ($57.00$–$142.00$ CTn), TVBN values ($6.72$–$33.60$ mg/100 g) and shear force values ($8.99$–$39.23$) were measured through traditional methods and compared with FTIR spectral data. Analysis of variance (ANOVA) was applied on data obtained through microbial and quality analyses and results revealed significant changes ($P < 0.05$) in the values of microbial load and quality parameters of chicken fillets during the storage. FTIR spectra were collected and PCA was applied to illuminate the wavenumbers potentially correlated to the spoilage of meat. PLS regression analysis permitted the estimates of microbial spoilage and quality parameters from the spectra with a fit of $R^2 = 0.66$ for TPC, $R^2 = 0.52$ for *Entetobacteriaceae* numbers and $R^2 = 0.56$ for TVBN analysis of stored broiler meat.

**Discussion**. PLS regression was applied for quantitative interpretation of spectra, which allowed estimates of microbial loads on chicken surfaces during the storage period. The

results suggest that FTIR spectra retain information regarding the spoilage of poultry meat.

**Conclusion**. The present work concluded that FTIR spectroscopy coupled with multivariate analysis can be successfully used for quantitative determination of poultry meat spoilage.

## INTRODUCTION

Meat is a very important part of the human diet due to the presence of several valuable nutrients such as proteins, vitamins and minerals. However, meat is also known as a highly perishable commodity due to its nutritional composition, which triggers biochemical changes responsible for spoilage. Several intrinsic and extrinsic factors are involved in the onset of meat spoilage, such as physical damage due to improper handling and storage conditions, unfavorable chemical changes caused by protein degradation and microbial activities (*Rahman et al., 2017*). Among these factors, microbial activity is considered as the major contributor of meat spoilage, resulting in the development of off-flavors, bad odors and slime which makes meat unfit for human consumption (*Rahman et al., 2017*). Thus, meat spoilage is considered as a subjective judgement by customers that can be affected by economic and cultural reflections and sensorial acuity of consumers (*Ammor, Argyri & Nychas, 2009*).

A vast range of methods has been used worldwide for detection of microbial spoilage and contamination of meat and meat products. Amongst these methods, the most extensively used approaches include organoleptic methods, physico-chemical analyses and cultural microbial techniques (*Herrero1 et al., 2017*; *Rahman et al., 2016*). These traditional methods used for detection of meat spoilage are quite time-consuming, labor intensive and need technical proficiency. Therefore, it is needed to introduce some rapid, cost effective, reagent-free and non-destructive methods to detect meat spoilage in an efficient way. In this regard, spectroscopic techniques have shown their potential for rapid and accurate prediction of microbial spoilage in meat and other food products (*Sahar & Dufour, 2014*). Accordingly, Fourier transform infrared (FTIR) spectroscopy can be used as a quick and non-invasive method for detecting meat spoilage. FTIR spectroscopy has shown its potential to predict biochemical changes in meat substrates and can be successfully employed to extract useful information about the muscle decomposition and metabolite generation due to the onset of spoilage (*Rahman et al., 2016*).

Several studies have shown the effectiveness of FTIR spectroscopy to detect the microbial spoilage of meat. For instance, *Ellis et al. (2002)* applied FTIR spectroscopy for rapid and on-line detection of microbial count on chicken breast fillets and revealed that FTIR has potential to be used for the determination of microbial safety and quality of meat and other food products during the processing and storage. Similarly, Ammor and

coworkers (*2009*) exploited FTIR spectroscopy along with chemometric analysis for non-destructive prediction of meat spoilage. They also analyzed microbial count, pH and sensorial attributes of beef subjected to various storage conditions and concluded that this technique can be successfully used for rapid and non-invasive monitoring of meat spoilage. Likewise, *Sahar & Dufour (2014)* used FTIR spectroscopy to predict bacterial spoilage of aerobically stored chicken breast fillets and concluded that this technique can be applied for on-line monitoring of microbial spoilage of meat. Additionally, *Grewal, Jaiswal & Jha (2015)* also used FTIR spectroscopy coupled with chemometric analysis for the detection of poultry meat specific bacteria and concluded that spectral windows in the regions of 4,000–575 $cm^{-1}$, 3,000–2,500 $cm^{-1}$ and 1,800–1,200 $cm^{-1}$ have the potential to classify poultry meat based on the presence of different pathogenic bacteria and level of contamination. Moreover, *Foca et al. (2016)* applied different spectral (Fourier transform mid infrared spectroscopy (FTMIR) and Fourier transform near infrared spectroscopy) and hyperspectral techniques for detection of lactic acid bacteria in sliced cooked ham and revealed that FTMIR spectroscopy in the region of 4,000–675 $cm^{-1}$ can be used in combination with multivariate analysis to get information regarding bacterial contamination in food samples. FTIR has also been used for the determination of molds in different food products (*Shapaval et al., 2017*). Accordingly, the current investigation was envisioned to investigate if FTIR spectra can be used for predicting the microbial load on meat surfaces and to explore the competence of FTIR spectroscopy in tandem with chemometric modeling to detect meat spoilage by predicting variations in microbial load, pH, color, texture and TVBN values on the surface of aerobically stored broiler breast fillets under refrigerated conditions.

## MATERIALS AND METHODS

### Procurement of materials

Broiler chicks (1.5–2.0 kg) were procured from the local market and slaughtered in the Meat Science and Technology Laboratory by following Halal Ethical Guidelines (*Department of Standards Malaysia, 2009*). Breast samples were separately packed into sterilized stripper bags for storage. Meanwhile other chemicals were also purchased from Sigma Aldrich (Darmstadt, Germany).

### Storage conditions

Meat samples were aerobically stored in polyethylene bags at 4 °C $\pm$ 0.5 °C (refrigeration temperature) and analyzed at regular intervals of 2 days (0, 2, 4, 6 and 8). Five separate samples were used at each storage day for analysis. The experiment was repeated twice & average values are used for statistical analysis.

### Microbial analysis

Nutrient agar medium was prepared for TPC by dissolving Nutrient Agar (2.8 g) in sterilized distilled water (100 mL) and autoclaved at 121 °C and 15 psi for 45 min. Likewise, MacConkey agar medium was prepared for *Enterobacteriaceae* members by dissolving MacConkey Agar (5.2 g) in sterilized distilled water (100 mL) and autoclaved

at 121 °C and 15 psi for 45 min. Media were put into petri dishes in Laminar Culture Hood and plates were left to solidify and then placed in an incubator at 37 °C for 24 h before use to confirm that the media were properly sterilized. Only those Petri dishes which did not observe any growth were selected. Surface microflora were collected from the chicken samples by using a sterile culture swab and transferred to the growth media for incubation (37 °C for 24 to 48 h). TPC and *Enterobacteriaceae* counts were counted using colony counter followed by visual analysis to observe morphology (pink colored spherical to oval shaped colonies for *E. coli*, colorless round-shaped colonies for *Salmonella*) and typical colony types. Actual microbial colony counts ($CFU/cm^2$) were then converted to logarithmic values (*Leblanc & Dufour, 2002*; *Sahar & Dufour, 2014*).

## Quality parameters
The pH (*Diaz et al., 2011*), CTn-color values (Color transmittance number) (*Rahman et al., 2017*), texture (*Piga et al., 2005*) and total volatile basic nitrogen contents (*Luo et al., 2011*) were measured by using their respective protocols. Afterwards, the obtained results were compared with spectral data through multivariate analysis for evaluating the potentiality of FTIR spectroscopy to predict meat spoilage.

## Statistical analysis
The experiment was conducted under Controlled Randomized Design (CRD) and analysis of variance (ANOVA) was applied for statistical interpretation of microbial and quality parameters using STATISTICS 8.1 software. Level of significance ($P < 0.05$) was measured by applying Fisher test through post-hoc analysis.

## Analysis on FTIR spectrophotometer
Mid Infrared (MIR) spectra were taken using ZnSe ATR (attenuated total reflectance) crystal in the range of 3,000 to 800 $cm^{-1}$ (resolution $= 4$ $cm^{-1}$) on FTIR spectrophotometer (Bruker Tensor 27) equipped with OPUS software using the method of *Sahar & Dufour (2014)* with slight modifications.

An average of 16 scans was taken each time, so 16 (No. of spectra on same sample) $\times$ 5 (total number of independent samples on each storage day) $\times$ 2 (Experiment was repeated twice on each sample) $= 160$ spectra for each storage time. $160 \times 5$ (storage interval) $= 800$ spectra in total were taken for this study. Additionally, reference spectra were also collected from clean crystal prior to run each sample. The crystal was cleaned after running each sample with ethanol and dried before running the next sample.

## Data acquisition
### Pre-treatment of FTIR spectra
For pre-treatment of FTIR spectra, baseline correction was applied by using The Unscrambler software (http://www.camo.com/rt/Products/Unscrambler/unscrambler.html).

### Principle component analysis (PCA)
PCA was applied on each offset of the standardized spectral data for drawing similarity maps to observe the similarities or differences among spectra and getting spectral patterns showing the most discriminant wavelengths. PCA is used to convert the large number of

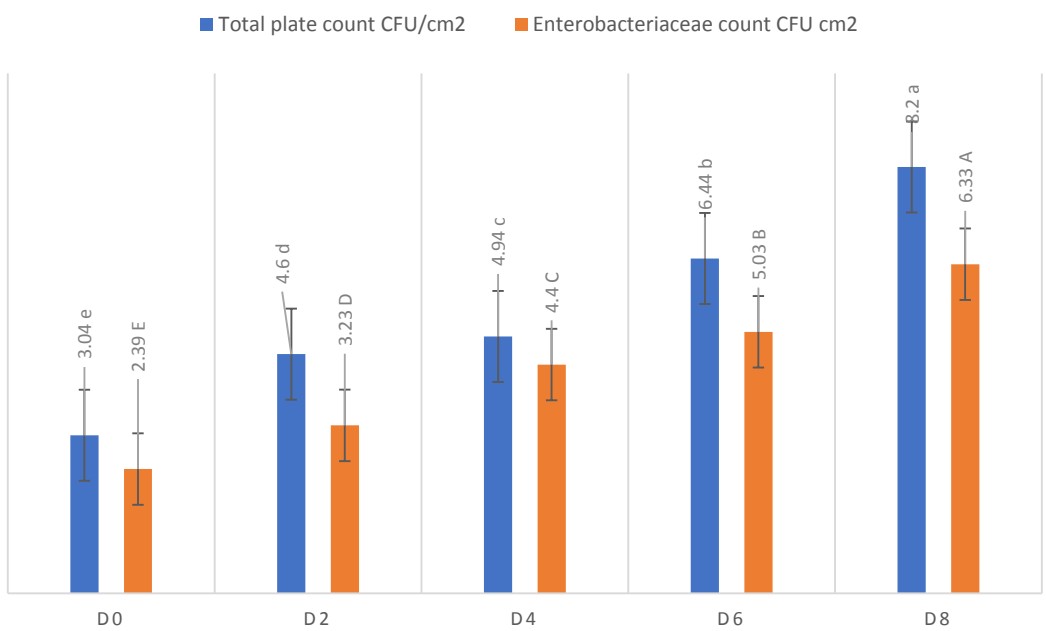

**Figure 1** **Microbial kinetics of poultry meat during refrigeration storage (Day 0 to Day 8).**

potentially correlated factors into small number of uncorrelated factors which are called principal components.

### PLS (Partial least square) regression analysis

PLS regression (*Vigneau et al., 2006*) was applied for cross-validation on random mode to observe fundamental relationship between two matrices i.e., FTIR spectral data and traditional microbial (TPC and *Enterobacteriaceae*) and quality evaluation (pH, color, shear force, TVBN) from broiler meat samples stored at refrigeration temperature. N-PLS regression was used to predict bacterial counts from the FTIR spectra because the variables to be predicted are characterized by a matrix (TPC and *Enterobacteriaceae* at the different storage times) instead of a vector (*Bro, 1996*). Root mean square error of calibration (RMSEC), root mean square of prediction (RMSEP) and coefficients of determination ($R^2$) of calibration and validation were also determined.

## RESULTS

### Microbial analysis through cultural method

A significant increase in the development of surface microflora was observed on meat samples during storage. Figure 1 depicts the results of TPC (log $CFU/cm^2$), which described that a considerable increase in TPC values was observed as storage time progressed. Lowest mean TPC value ($3.04 \pm 0.05$ log $CFU/cm^2$) was chronicled on the first day of storage and was increased in a significant manner ($8.20 \pm 0.01$ log $CFU/cm^2$) up to day 8 which evidently presented the onset of poultry meat spoilage during storage. Figure 1 also illustrats that the recorded logarithmic value of *Enterobacteriaceae* family was $2.39 \pm 0.01$ log $CFU/cm^2$

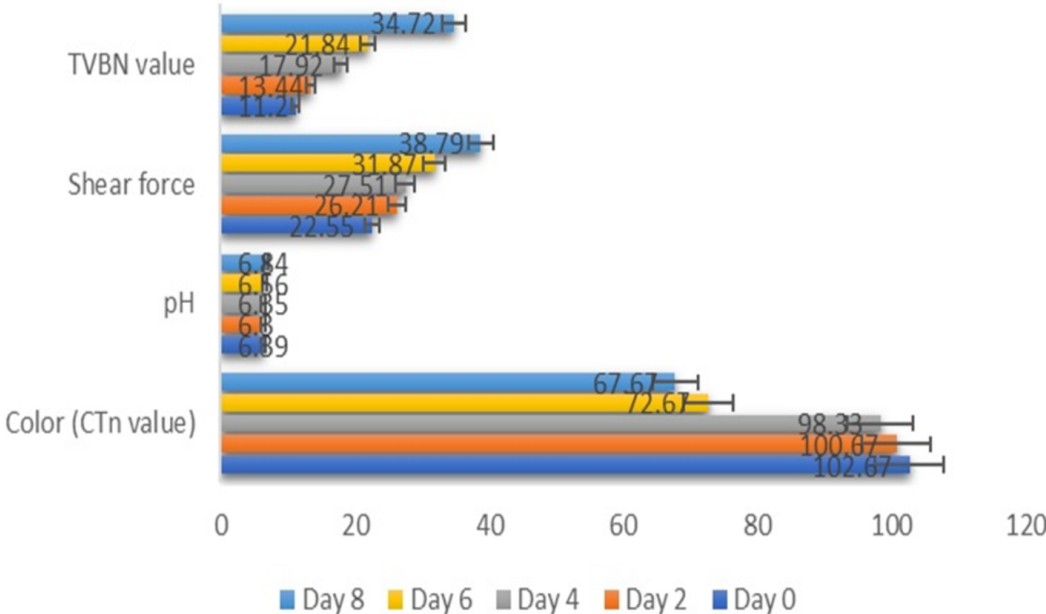

**Figure 2 Quality parameters.** Variations in quality parameters of poultry meat during refrigeration storage (Day 0 to Day 8).

on Day 0 which was increased up to $6.33 \pm 0.01$ log CFU/cm$^2$ at the termination of trial (Day 8).

## Determining meat quality parameters through traditional methods

Different meat quality parameters viz. pH, color (CTn value), shear force and TVBN values were determined by using traditional methods to measure the spoilage level of aerobically stored chicken fillets during the storage time at constant interval of two days (Fig. 2). Results regarding TVBN analysis depicted a momentous increase in TVBN values during the storage period (11.20 to 34.72 mg/100 g) which clearly indicated the spoilage of meat. Similarly, shear force values were also increased in a significant way from 22.55 to 34.72 during the storage period. However, non-significant variations were observed in pH values of aerobically stored chicken breast fillets. The change in color values from 102.67 (CTn) to 67.67 (CTn) was indicative of meat spoilage during the storage.

## FTIR spectral interpretation

Meat samples were also analyzed through FTIR spectrophotometer in the mid infrared range and peaks were obtained ranging from 3,000 to 800 cm$^{-1}$. The following graph (Fig. 3) presents variations in the peak absorbance analyzed through FTIR at various stages of storage (day 0 and day 8). A significant increase in the absorbance of different bands was also observed between spectra of meat samples from different storage intervals. Deviations in the spectral results have been illustrated in Fig. 3.

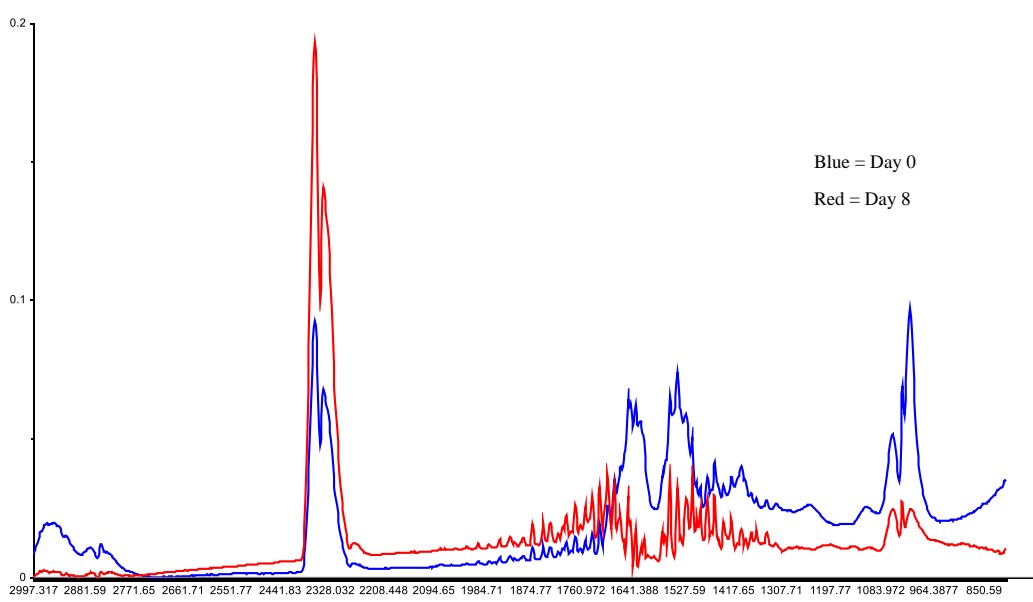

**Figure 3** **FTIR spectra.** Comparison of FTIR spectra collected from refrigerated chicken samples at Day 0 and Day 8.

## Principle component analysis (PCA)

The similarity maps developed by applying PCA on FTIR spectra collected from stored meat samples at different days (0, 2, 4, 6, 8) revealed that PC1 and PC2 accounted for 62% of the total variance (Fig. 4). Additionally, the classification of meat spoilage during refrigeration storage presented that PC1 and PC2 predicted 36% and 26% of the total variance respectively. Moreover, PCA for describing the relationship for variations among various quality attributes due to microbial spoilage of meat showed that first two components completely described the total variance (PC1 = 88%, PC2 = 12%) as shown in Fig. 5. The findings revealed that these PCs explained the variance of spectral data at specific range of wavenumbers (1,750–1,200 cm$^{-1}$) describing different biochemical changes occurred in the meat samples during the spoilage process. These wavenumbers mainly corresponded to the absorption of amide I and amide II bands due to C-N bond stretching, fatty acids (CH$_2$ bond scissoring) and amines (C-N stretching) but the major variance of spectral data set was explained by amide I & II bands and amine groups (*Boubellouta & Dufour, 2012*).

## PLS regression for predicting bacterial load and quality of chicken fillets

Graphical representations of regression plots between predicted and reference values of total plate count, *Enterobacteriaceae* count and TVBN values are shown in Figs. 6A, 6B and 7. In these graphs, the predicted values are illustrated by red dots while reference points are shown in blue color. A good distribution of samples around the lines of equity can
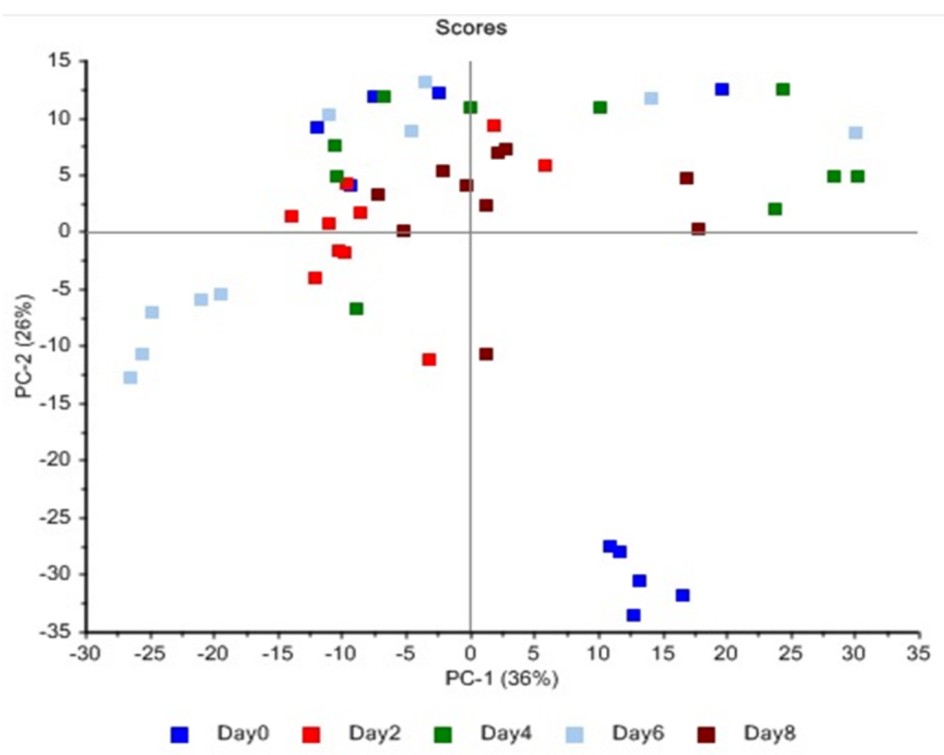

**Figure 4** PCA similarity maps defined by PC1 and PC2 recorded during storage period (Day 0–8).

be shown in these graphs. The dots which are placed away from the central line show the spoilage of samples due to an increase in the microbial load during the storage.

Mean values, coefficients of determination ($R^2$), standard deviations, root mean square errors of prediction (RMSEP), ratio performance deviations (RPD), standard errors for calibration (SEC) and coefficients of determination of cross-validation (1-VR) for TPC, *Enterobacteriaceae* count, color (CTn values), pH and TVBN values of meat samples are depicted in Table 1. The statistical description of TPC and *Enterobacteriaceae* counts reported the mean values of 4.02 CFU/cm$^2$ and 3.47 CFU/cm$^2$ correspondingly. Additionally, the mean values for color, pH, TVBN and shear force were 92.10 CTn, 5.81, 14.46 and 23.36 mg/100 g respectively.

Prediction models designed from selected equations to predict microbial load and other spoilage indicators in chicken breast fillets determined moderate accuracy for predictions of TPC ($R^2C = 0.77$, $R^2V = 0.66$, RMSEP = 0.75, RPD = 1.24, SEC = 0.59), *Enterobacteriaceae* members ($R^2C = 0.70$, $R^2V = 0.52$, RMSEP = 0.75, RPD = 1.13, SEC = 0.58), color ($R^2C = 0.65$, $R^2V = 0.33$, RMSEP = 21.02, RPD = 0.97, SEC = 15.17), pH ($R^2 = 0.21$, RMSEP = 0.79, RPD = 1.00, SEC = 0.69), TVBN ($R^2C = 0.74$, $R^2V = 0.56$ RMSEP = 3.19, RPD = 1.27, SEC = 2.34) and shear force ($R^2C = 0.50$, $R^2V = 0.34$, RMSEP = 5.72, RPD = 1.19, SEC = 4.75).

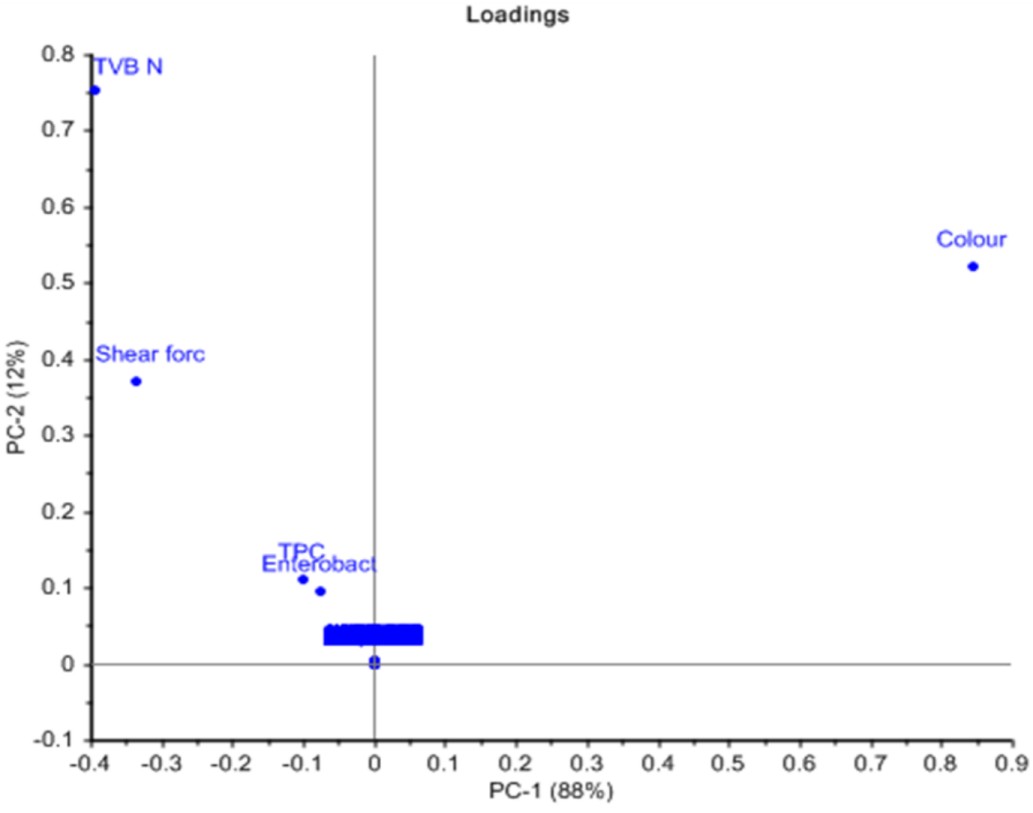

**Figure 5** PCA plot defined by PC1 and PC2 for microbial and quality parameters of poultry meat during storage.

## DISCUSSION

### Determining of microbial load through traditional method

Results of cultural techniques to identify pathogenic microorganisms on the surface of aerobically stored broiler fillets showed that initial *Enterobacteriaceae* load was increased on the surface of the meat with the progression of storage time. The onset of meat spoilage generally occurs due to the metabolic activity of different microbial species introduced after slaughtering. Several factors are involved in defining the presence of microbes on the surface of the meat. The major contributors include environmental conditions, packaging type, initial microbial load and the propagation ability of microorganisms. The findings of the present work have revealed that the TPC and *Enterobacteriaceae* count of raw chicken fillets were significantly affected in a direct manner by storage interval (*Balamatsia et al., 2006*).

### Determining meat quality parameters through traditional methods

Results obtained from TVBN analysis of aerobically stored broiler meat fillets revealed a significant increase in TVBN values with the progression of time, which clearly depicted the spoilage due to the loss of volatile nitrogen during the storage period, resulting from putrefaction of proteins from microbial and/or enzymatic activities. Similarly, the

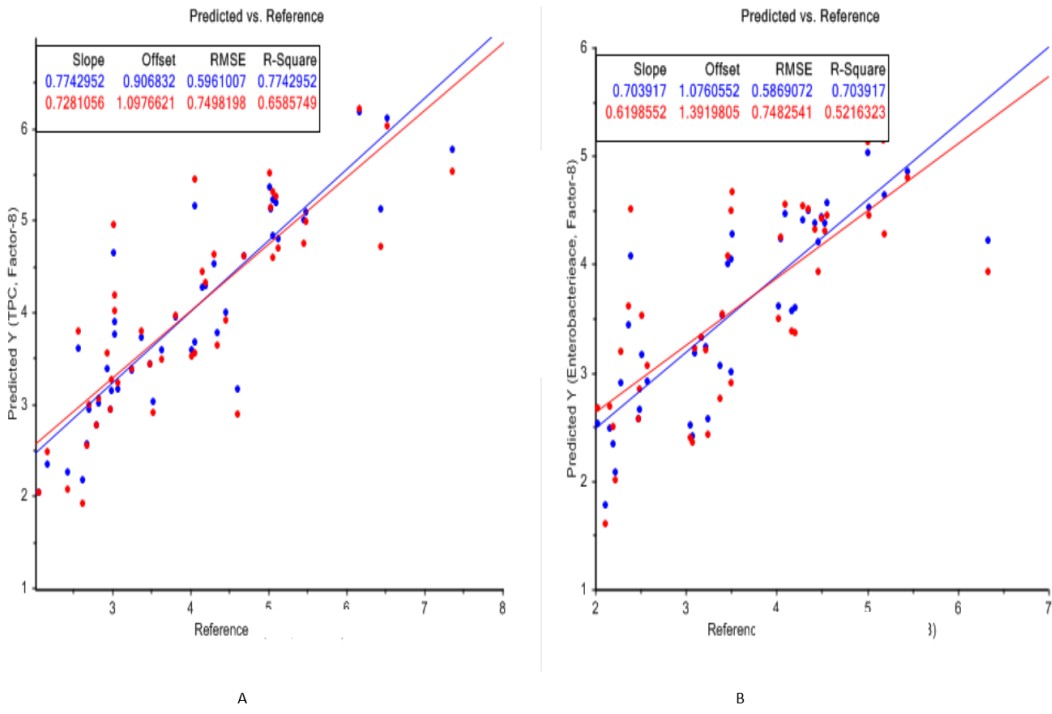

**Figure 6 PLS regression plots (reference vs. prediction) of FTIR spectra for total plate count (6A) and *Enterobacteriaceae* count (6B) of chicken fillets during storage.**

increase in shear-force values was due to the hardening of muscle fibers during storage. Additionally, inferences about color (CTn value) analysis depicted expressive variations in the color values during storage. Decreased CTn values demonstrated a darker color in meat due to microbial spoilage. The findings of the current investigation are in accordance with the work of *Rahman et al. (2017)* who investigated the impact of various antimicrobial agents on different quality parameters of poultry meat and reported similar variations in the above-mentioned parameters.

## Explanation of FTIR spectra

The spectral results showed decreased absorbance of peaks at 1,650 cm$^{-1}$ during the refrigerated storage of chicken breast fillets. Variations in the absorbance and position of peaks collected from broiler meat samples during the storage are prescribed by the changes in the band stretching of C-H, O-H, N-N and N-H functional groups.

## Principle component analysis (PCA)

PCA was applied on the large dataset to reduce its multidimensionality. The technique is also useful in identifying natural clusters in the data set in which the first principal component (PC1) shows the largest level of variation followed by the second component (PC2) which is useful in describing the second most important factor of remaining analysis and so on (*Sahar & Dufour, 2014*). In this study, the score plots obtained from PCA are useful for interpreting the similarities and differences between the growth rate of bacteria and storage

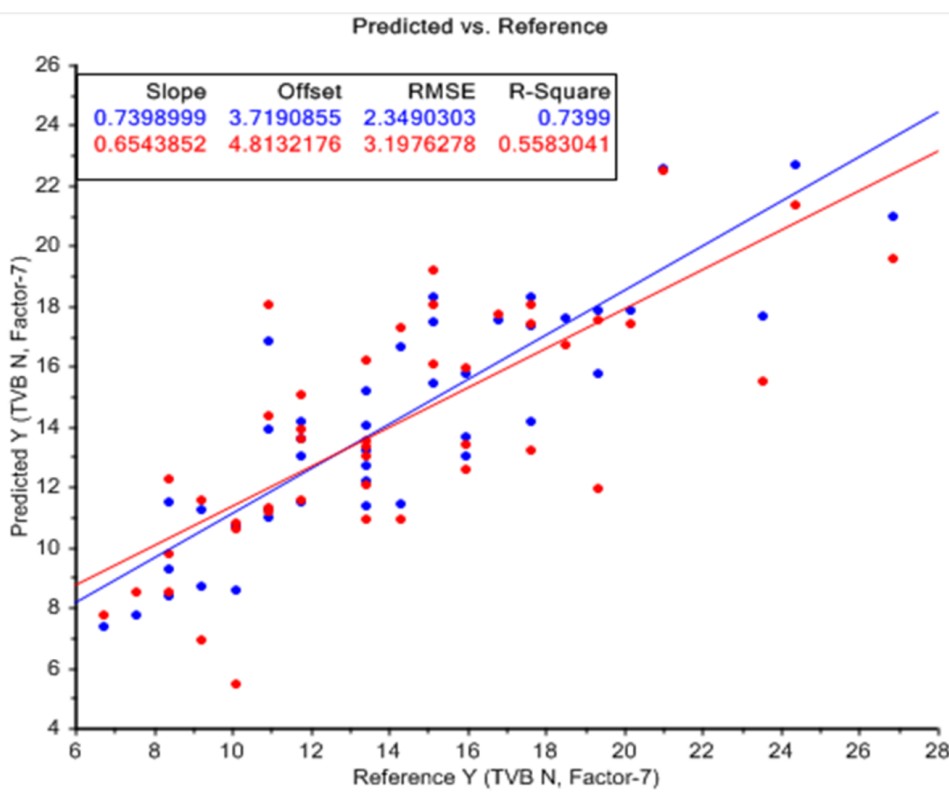

**Figure 7** **PLS regression plot.** Reference vs. prediction plot for TVBN analysis of broiler meat stored under refrigerated conditions.

**Table 1** Results of PLS regression carried out on FTIR spectra (3,000–800 cm$^{-1}$) and meat spoilage parameters obtained by traditional techniques on chicken samples at $4 \pm 2\,°C$.

| Parameter | $R^2C$ | SEC | 1-VR | RMSEP | Mean | SD | RPD |
|---|---|---|---|---|---|---|---|
| TPC | 0.77 | 0.59 | 0.66 | 0.75 | 4.02 | 1.40 | 1.24 |
| Entero | 0.70 | 0.58 | 0.52 | 0.75 | 3.47 | 1.13 | 1.13 |
| Color | 0.65 | 15.17 | 0.33 | 21.02 | 92.10 | 28.02 | 0.97 |
| pH | 0.21 | 0.69 | 0.03 | 0.79 | 5.81 | 0.79 | 1.00 |
| TVBN | 0.74 | 2.34 | 0.56 | 3.19 | 14.46 | 5.43 | 1.27 |
| Shear force | 0.50 | 4.75 | 0.35 | 5.72 | 23.36 | 6.79 | 1.19 |

time. The closeness of the analyzed samples in the score plot determines the similarity of samples with respect to the evaluated principal component score. The outcomes of PCA applied on different data sets provided information concerning the discernment of various samples. The results obtained from PCA also highlighted the effectiveness of FTIR as a reliable and comprehensive approach for the grouping of microbes on broiler meat as a function of storage time. Additionally, the classification of meat spoilage during refrigeration storage presented that PC2, which is responsible for 26% of the variance, separated the day 0 samples from the rest of the stored meat samples. Similarly, the second

PCA graph clearly showed that TPC and *Enterobacteriaceae* count are mainly responsible for this grouping. The microbial load was at a minimum in day 0 samples, so these are grouped separately by PCA plot. The findings of the current investigation have revealed that FTIR spectroscopy has the potential to identify the spoilage of chicken fillets due to microbial activities. Moreover, PCA results also provided useful information about various biochemical changes in meat composition because of microbial spoilage.

### PLS regression models

Better prediction inferences were observed for all the parameters of broiler meat samples during refrigeration storage. The results of the present work showed that even though the current model showed less accuracy than the model developed by *Sahar & Dufour (2014)*, it provided satisfying results for the potentiality of FTIR spectroscopy in predicting meat spoilage. Additionally, the problem of developing average models can be overcome by broadening the dataset and by predicting the spoilage at different storage temperatures.

## CONCLUSION

The present investigation concluded that FTIR spectroscopy can be used to extract useful information regarding meat spoilage during storage. Results of PLS regression revealed that satisfactory prediction of meat spoilage is possible even with a small number of PLS factors. Subsequently, more research work is needed with a large sample size for establishing the utility of FTIR spectroscopy for prediction of meat spoilage. Additionally, the authors are also doing work on exploring the role of FTIR and other spectroscopic techniques for identifying the individual bacterial species in complex food matrices and detecting other meat quality, safety and authenticity parameters.

## ACKNOWLEDGEMENTS

The authors are gratified to NIFSAT & IM, UAF, for allowing work in their laboratories.

### Funding

This research was supported by the Higher Education Commission, Pakistan. The funders had no role in study design, data collection and analysis, decision to publish, or preparation of the manuscript.

### Grant Disclosures

The following grant information was disclosed by the authors:
Higher Education Commission Pakistan.

### Competing Interests

The author declare there are no competing interests.

## Author Contributions

- Ubaid ur Rahman conceived and designed the experiments, performed the experiments, analyzed the data, prepared figures and/or tables, authored or reviewed drafts of the paper.
- Amna Sahar conceived and designed the experiments, analyzed the data, contributed reagents/materials/analysis tools, prepared figures and/or tables, authored or reviewed drafts of the paper, approved the final draft.
- Imran Pasha contributed reagents/materials/analysis tools, authored or reviewed drafts of the paper, approved the final draft.
- Sajjad ur Rahman conceived and designed the experiments, contributed reagents/-materials/analysis tools, authored or reviewed drafts of the paper, approved the final draft.
- Anum Ishaq performed the experiments, analyzed the data, prepared figures and/or tables, authored or reviewed drafts of the paper.

## Data Availability

The raw data are provided in the Data S1.

## Supplemental Information

Supplemental information for this article can be found online at http://dx.doi.org/10.7717/peerj.5376#supplemental-information.

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
