# Peer review of "Assessing the capability of Fourier transform infrared spectroscopy in tandem with chemometric analysis for predicting poultry meat spoilage"

_PeerJ, doi:10.7717/peerj.5376_

## Round 0.1 · original submission · Major Revisions

The basic idea here is very good. Infrared spectroscopy shows great promise for characterizing food spoilage quickly and easily. However, as can be seen in the reviewer comments, there is much room for improvement in the manuscript.

Please take all of the reviewer comments very seriously in your revision. Furthermore, there is a need for more methodological detail and better referencing throughout the manuscript. You can use papers like "Rapid monitoring of the spoilage of minced beef stored under conventionally and active packaging conditions using Fourier transform infrared spectroscopy in tandem with chemometrics" from Meat Science as a model.

Make certain your statistics are appropriate. For instance, should you be using a repeated measures ANOVA? I would also strongly suggest doing a plot of predicted versus reference bacterial account, as is typical for chemometric analyses, as it really is the bottom line. You want to see if you can predict food spoilage using FTIR, so it is such a plot that will be of central concern to readers. Lastly, make certain you are specific as to how you cross-validated your model. I am also worried that your baseline corrections may not have worked given your figure with the spectra. Please double check that (and see the comparable figure from the paper above).

Reviewer 1 ·

Basic reporting

- English is ok
- Extremely poor sue of literature and very weak knowledge of the background is provided
- this is ok
- yes but this is not the problem

Experimental design

- this is ok
- can be considered as such
- limited
- this is ok

Validity of the findings

- The experimental design is weak (single T, and single analysis was performed) and thus not solid conclusion can be drawn.
In this field numerous of studies have been published with a greater amount of data; thus the limited data that have been used within this study do not allow to support the conclusions reported in this article. The analysis (PCA,PLS) have been performed with inadequate number of samples; this is in my opinion a main concern regarding the impact of this study
In other words this study can be considered as VERY preliminary and more data are needed.

Additional comments

Further experimental work is needed before this study can be considered as solid. The limited number of samples examined are not enough (4 points for microbiological analysis and 4 points for FTIR). The absent of any information regarding those microorganisms that are the main spoilers (pseudomonads) are also a weak point.

Reviewer 2 ·

Basic reporting

Suggestion to rewrite the abstract considering the following:
The abstract structure includes copies of same text in both methods and results without distinction between the two sections.
Line 29: what were the statistics for this significance and what is the direction and strength of correlation
Lines 30-31: PCA does not measure strength of models
Line 31: add prediction stats and avoid the subjective "good predictions"
results relating Fourier transform infrared spectroscopy and traditional methods are not presented in the abstract
Introduction
The objectives of the study are not clear. The authors quote a similar study by Ammor et al. (2009) exploiting FTIR spectroscopy 55 along with chemometric analysis for non-destructive prediction of meat spoilage. Does this study aim to repeat these findings but using other spoilage measures?

Experimental design

What was the sample size?
was there replications of the experiments?
What is the citation for Halal Ethical Guidelines? Are these standard scientific protocols?
Lines 75-76: which media and which incubation conditions- what morphology and typical colony types categories?
Lines 96-97: it is not cleae if PCA or is it discriminant analysis is used.

Validity of the findings

Lines 134-136: value of FTIR spectra from isolated pure cultures of different members of Enterobacteriaceae is not clear as isolated cultures would not show differences in meat quality
Lines 141-148: the principal components have not been clearly explained and what is implied by the various groupings of the variables
Lines 200-205: it is not clear from PCA results how these findings are supported by this research and these may not be valid findings from the results unless this is made clear.
What is the sensitivity and specificity of this new approach.

Additional comments

Line 75: change "Total plate counts (TPC) Enterobacteriaceae" to "Total plate counts (TPC) and Enterobacteriaceae"
Line 100: Is this Partial least squares regression? Should be written in full at place of first mention
Lines 174-179: these do not really imply new findings as it is quiet general knowledge that bacteria multiply in foods such as meat over time
Line 216: do the authors mean "promoting" rather than "glorifying"? use different choice of wording.

---

## Round 0.2 · Minor Revisions

The manuscript is improved. There are still lots of small, and not so small, language issues, and the referencing continues to be too light. If you email I can send a few more specific suggestions as a track changes word document.

---

## Round 0.3 · accepted · Accept

The English language is still in less than stellar shape. That said, a good faith effort has been made so I'm inclined to accept at this point.

#